# “Dietitians May Only Have One Chance”—The Realities of Treating Obesity in Private Practice in Australia

**DOI:** 10.3390/healthcare10020404

**Published:** 2022-02-21

**Authors:** Claudia Harper, Radhika V. Seimon, Amanda Sainsbury, Judith Maher

**Affiliations:** 1The Boden Collaboration for Obesity, Nutrition, Exercise, and Eating Disorders, Charles Perkins Centre, Faculty of Medicine and Health, The University of Sydney, Camperdown, Sydney, NSW 2040, Australia; radhika.seimon@health.nsw.gov.au; 2School of Human Sciences, The University of Western Australia, Crawley, WA 2009, Australia; amanda.salis@uwa.edu.au; 3Faculty of Science, Health, Education and Engineering, School of Health and Sport Sciences, University of the Sunshine Coast, Sippy Downs, QLD 4556, Australia; jmaher@usc.edu.au

**Keywords:** VLED, qualitative, obesity, weight loss, private practice

## Abstract

Introduction: Overweight and obesity are the leading contributors to non-fatal burden of disease in Australia. Very low energy diets (VLEDs) comprising of meal replacement products (MRP) effectively induce substantial weight loss in people with obesity, yet they are rarely used as a first line treatment. Dietitians in private practice are perfectly placed to administer treatments for obesity; however, little is known about the preferred interventions used or their attitudes to incorporating VLEDs and MRPs into their treatments for overweight and obesity. Methods: This study used descriptive qualitative methods to explore accredited practicing dietitians’ (APDs’) perspectives and practices regarding obesity and obesity interventions, including the use of VLEDs and MRPs. Qualitative in-depth semi-structured interviews were conducted with 20 dietitians who had experience in private practice and in treating obesity. Transcribed interviews were analysed thematically using the technique of template analysis. Results: In the context within which dietitians’ practice was found to be a barrier to using evidence-based practice (EBP) for obesity treatment, four overarching themes were found. These were: (1) patient-centred care is the dietitians’ preferred intervention model; (2) VLEDs promote weight loss in specific situations; (3) systemic barriers constrain effective dietetic practice and equitable access to all, and (4) successful outcomes are predicated on working outside of systemic barriers. Conclusion: Dietitians in private practice are well placed and able to provide life-enhancing and evidence-based treatments for overweight and obesity and associated chronic disease in the community. However, systemic barriers need to be addressed to provide equitable access to effective care irrespective of socio-economic status.

## 1. Introduction

The prevalence of overweight and obesity has reached epidemic proportions around the world [1,2]. Obesity-associated co-morbidities and healthcare costs are ever increasing [2,3]. In Australia, data from the 2017–2018 Australian Bureau of Statistics’ National Health Survey showed that two thirds (67.0%) of Australian adults have overweight or obesity (12.5 million people), an increase from 63.4% in 2014–2015 [4]. Overweight or obesity is a known risk factor for 22 high-cost diseases including diabetes, musculoskeletal conditions (e.g., osteoarthritis and back pain), cardiovascular disease, kidney disease, sleep apnoea, asthma, dementia and various cancers [4,5,6]. Moderate weight loss is known to reduce blood pressure, Type 2 diabetes biomarkers, circulating lipids, and other CVD risk biomarkers, as well as, sleep apnoea and hip/knee osteoarthritis [7].

To combat rising chronic disease, secondary to overweight and obesity, the Australian government provides Medicare rebates under the Medicare Benefits Scheme (MBS). Chronic disease management (CDM) rebate items, previously known as Enhanced Primary Care (EPC) items, provide people with chronic illness up to five 20 min consultations with allied health professionals over the course of a year to provide extended care [8]. Consultations are funded by the Australian Government as a rebate. A practitioner can choose to bulk bill, where the rebate covers the full cost of consultation or charge a gap fee (a fee over and above the Medicare rebate amount) for their service. The General Practitioner (GP) is the gate keeper to these pathways for clinical treatment, and determines how the five Medicare-eligible allied health services are allocated (Appendix A) [8,9]. For example, someone may get 1 × dietitian, 2 × podiatrist and 2 × exercise physiologist appointments, depending on their needs. This extended care, alongside medical care, attempts to treat the symptoms and causes of chronic disease. As obesity is a risk factor and underlying cause of many chronic diseases, the initial treatment goal in healthcare is often achieving a clinically significant weight loss. Long term weight loss of 5% from original weight is considered the threshold of weight loss for clinical significance [10]. However, certain risk factors, such as glycaemic measures and triglycerides, may improve with as little as 3% weight loss [10]. Additionally, it has been shown that greater weight loss produces better health benefits [10].

Meta-analyses indicate that the most effective long-term non-surgical, non-pharmacological individual-level weight loss treatments are very low energy diets (VLEDs) when administered under medical supervision [11,12]. VLEDs provide less tan 3350 kJ/800 kcal per day and are achieved primarily by total Meal Replacement Products (MRPs), where all meals and snacks are replaced with shakes or soups. The greater initial weight loss after VLEDs compared to other diets has been shown to be predictive of long-term maintenance of a lower body weight [13,14,15]. Those who remain engaged with support after a VLED can retain clinically significant weight loss in the long term (3 years) [16]. In a recent randomised controlled trial, participants who underwent a 16-week VLED, compared to participants on a conventional food-based diet were 2.6 times more likely (42% versus 16%) to have lost 10% or more of their initial body weight at a 3 year follow up [17]. In a systematic review of qualitative experiences, findings showed VLEDs were easy to follow, required minimal decisions about food and rapid initial weight loss motivated adherence. However, some participants found the diet anti-social and experienced minor negative side effects (e.g., feeling cold, constipation) [18].

Despite the effectiveness and reported acceptability of VLEDs, health professionals tend to use them only as part of the prerequisites for surgery, or as a last resort [19,20,21]. VLEDs are extensively used, from two to six weeks, prior to bariatric and knee replacement surgery and effectively induce fast weight loss and liver shrinkage in people with obesity [19,21,22,23] Dietitians Australia (DA) (formerly Dietitians Association of Australia (DAA)) best practice guidelines state that there is Level A evidence (i.e., a body of evidence that can be trusted to guide practice) that incorporating meal replacements into a weight loss diet, monitored by a health professional, provides greater weight loss in adults with overweight or obesity than general dietary advice [24]. Previous studies show few Australian dietitians (3.2%) would use a VLED for obesity treatment or actually prescribe VLEDs in practice (1.5%) [25,26] More recently, a survey of 197 health care practitioners (HCPs) working in weight management across Australia, including dietitians, reported prescribing a diet containing MRPs to a median 7% of clients seeking weight loss treatment [27]. Primary reasons for low use were safety concerns, concerns about promoting weight cycling, and perceived potential negative psychological impact [27]. Regardless of the mode of weight loss chosen, interventions are more successful with frequent contact in the first year with a trained interventionist [28]. Weight regain is less in those that continue with at least bimonthly or more frequent contact after the first year of a comprehensive lifestyle intervention [28]. In Australia, this contact is often via the primary healthcare system.

Private practicing (primary care) dietitians are part of the Primary Health Care network [29,30]. In 2017, there were 1944 private practicing dietitians out of 5360 credentialled dietitians in Australia [31]. Primary health care is situated outside of hospital and government sectors and provides front line health care services in the community [29,30] Dietitians are evidence-based practitioners trained to implement individualised Medical Nutrition Therapy and in a position to support effective treatments across a broad range of health issues including obesity and chronic diseases [32]. Dietetic practitioner reticence to use VLEDs in practice warrants investigation. As such, the aim of this study was to use qualitative interviews to explore in-depth, accredited practicing dietitians’ current attitudes to obesity management including the use of VLEDs and MRPs, and the perceived barriers in care with regards to treating overweight and obesity in private practice.

## 2. Methods

### 2.1. Study Design

This study used a qualitative descriptive research design to capture participants’ experiences using everyday language [33]. In this case, private practising Dietitians’ experiences of treating overweight and obesity and their use of MRPs were explored via semi-structured interviews. Qualitative description allows the researcher to explore a specified area without forcing a priori theories onto the exploratory or analytical processes and present findings of the phenomena in a straightforward and comprehensive way [34]. It is a priority to stay close to the data and relay the findings as descriptions. Here themes are descriptive statements, to allow the reader insight into the participant experiences [34]. This study was approved by the Sydney Local Health District Ethics Committee (Royal Prince Alfred Hospital Zone, X16-0316 & HREC/16/RPAH/420). All participants were provided with information and gave written informed consent prior to participation.

### 2.2. Participant Recruitment

Invitation emails (Appendix B) were sent to practicing dietitians identified via online searches. A total of 64 emails were sent over a period of 8 months between August 2018 and April 2019, using email addresses identified in this way. A total of 31 dietitians replied and 20 were subsequently interviewed. The remaining 11 replied that they were interested in the study subject but did not have time to participate. An invitation advertisement targeted towards members of the DA was also paid for ($49.00) and posted in relevant DA interest groups, and a 50-word invite was distributed to in the member update email. (Appendix C). No dietitians were recruited via this method.

### 2.3. Data Collection

Semi-structured interviews were held in person, via Zoom or over the telephone, depending on the participant’s preference. Individual interviews ranged from 45 min to 1 h and 50 min in duration, for a total of 24 h and 57 min for all interviews combined, and were audio recorded. An interview guide (Appendix D) was developed through discussions between researchers that included dietitians (CH, JM), researchers experienced in VLEDs (AS, RVS) and qualitative research methodology (CH, JM). The interview guide gave a general structure to the data collection and allowed the researcher to explore areas not considered at conception of the guide. The interviews were conducted by a PhD candidate and dietitian (CH) in the period between June 2019 and February 2020. Data collection ceased when no new major concepts around the topic were being discussed in the interviews and the researchers felt that the data supplied enough content to gain an in-depth understanding of the research questions [35].

### 2.4. Data Analysis

The interviews were transcribed verbatim by hand (CH). The process of transcribing allowed the researcher to engage with the data, and brief notes were made to aid the initial analysis. Initial analysis was undertaken in NVivo 12 (QSR International Pty Ltd., Burlington, MA, USA, Version 12, 2019), by hand. Interviews were firstly analysed using line-by-line coding, allowing the researchers to examine the experiences and perceptions as described by participants [36]. Common concepts across the data and connections between concepts were identified. Due to the system-related experiences of dietitians in the sample, possible theories, such as systems theory and normalisation process theory, were examined for the possibility of aiding in the description of these experiences. However, it was found that dietitians in private practice have unique elements that are difficult to define using existing theories. Subsequently a coding framework was built in MS Word that incorporated the research aims and the emerging concepts in the data. When using template analysis it is usual to produce an initial version of the template on the basis of a sub-set of the data [37]. In practice, a table of columns was built with each column representing preliminary identified concepts with extra columns for concepts that were not yet identified but may prove interesting, related or crucial to the preliminary codes already found during initial coding in NVIVO. The whole data set was re-analysed (through another pass) as concepts were fit into the table, put into the extra columns or discarded. The template analysis technique was applied to organise the preliminary codes into discreet categories that were previously identified in NVIVO [37]. These categories were further analysed to determine overarching common themes that would provide a straightforward descriptive summary of the phenomena that the dietitians in this study were depicting [38]. Once the broad themes had been developed, the researchers returned to the raw data within each theme to check that the actual experiences of the participants were well reflected in the overarching themes. At all stages of this analysis, the main researcher (CH) discussed emerging categories and themes with experienced health researchers (AS, RVS) and a qualitative researcher (JM). A COREQ checklist was completed (Appendix E) [39].

## 3. Results

### 3.1. Participants

This study included 20 dietitians who had previous or current experience in dietetic practice in Australia across several areas (Table 1). The length of time that participants had spent working as a dietitian in private practice spanned 4 to 30 years. Two dietitians included in the sample were male. The dietitians in this study were a mixture of specialist and generalist, but all had experienced working with obesity and chronic disease in some capacity. All 20 dietitians participating in this study were either self-employed, in a sub-contracting relationship with a general practice or had previous experience working in this capacity. They also reported having varying experiences throughout dietetic practice in the public health system, research, bariatric surgery clinics, community health and private practice (Table 1). All dietitians in this study had some experience with Chronic Disease Management (CDM)/Enhanced primary care (EPC) referrals (Government-funded Medicare benefit scheme for patients to access allied health care for chronic disease management) via General Practitioners (GP). When patients accessed dietetics through these Medicare initiatives via GP referrals, weight loss was one of the main interventions used to address the chronic health disease listed on the referral. All participants expressed that it was rare to see a patient with uncomplicated overweight or obesity and that often, whilst obesity was a risk factor for their condition, it was not the primary reason for the patient seeking help. This is because Australia does not recognise obesity as a chronic disease, but instead lists it as a risk factor and, therefore, obesity in and of itself, does not qualify for the Medicare Benefit Scheme [40].

### 3.2. Overview of Results

There were four major descriptive themes found during the analysis of the dietitians’ interviews in this study, with several sub themes (Table 2). The themes found in this study showed that 1 *Patient-centred care is the dietitians’ preferred intervention model* and that Dietitians report 2 *VLEDs promote weight loss and motivation*. 2.1 *Experience-informed VLED use* and 2.2 *VLED require long term involvement.* Additionally, this research also exposed that there are three major *systemic barriers that constrain effective dietetic practice*. These barriers prevented the dietitians in this study from effectively delivering Evidence Based Practice (EBP) in a large proportion of their clients. It was found that the 3.1 *Medicare and CDM scheme limits effective dietetic care*. Within the Medicare and CDM scheme, dietitians described that 3.1.1 *insufficient rebate through Medicare could cause cutting corners*, the 3.1.2 *length of consult is insufficient to adequately address chronic disease* and the 3.1.3 *number of visits through Medicare does not ensure adequate follow up*. Another barrier common in private practice was 3.2 *poor access to health information makes assessment and treatment goals difficult*. Lastly, when dietitians where successful in helping patients to reduce their obesity and improve their chronic disease states, 4 *successful outcomes are predicated on working outside of systemic barriers.* We outline these four major themes and related subthemes in detail below and provide additional quotations in Appendix F.

#### 3.2.1. Patient-Centred Care Is Dietitians’ Preferred Intervention Model

Dietitians in this study reported that, first and foremost, their preferred approach towards helping clients with their health is a patient-centred care model. This entailed taking a detailed and thorough patient history, building rapport, and using a collaborative decision-making model of care. Unless, the type of diet was pre-designated, for example, pre-surgery or online programme, dietitians most often would favour small patient-specific lifestyle changes over time.


*D14 “It’s certainly not one size fits all. […] I think that for me, the most important thing (is) to have a good rapport with your patient, to let them know that you are walking the walk with them, that you’re not just sending them away with a regime.”*



*D13 “I try and get them to put in as much of the information as possible to try and find something that will work for them. […] So, I try and set goals with the patient that are almost set by them in a way.”*


#### 3.2.2. VLEDs Promote Weight Loss and Motivation

The dietitians reported they would often offer VLEDs as an intervention option for their patient, or a referring GP might specify a VLED for their patient. It was also common for patients themselves to request help in following a VLED. Two dietitians also sold MRPs, that are only available via a trained consultant, through their practice. A common reason to offer a VLED was to kick start weight loss and provide motivation. The dietitians in this sample felt that MRPs used in an intensive VLED, or as part of a reduced energy diet, had an important place in the dietitians ‘tool box’. They were more likely to use these to replace one or two meals instead of a more intensive VLED. Many had devised diets that utilised meal replacement products as part of a healthy eating plan to induce an energy deficit that fits in with the client’s lifestyle more easily. One dietitian (who had no experience with VLEDs) was opposed to their use, for ideological reasons.


*D16 “I probably have a 2 pronged approach. One is to just start that weight loss to get some motivation happening with people if they’ve constantly been failing, failing, failing to get a bit of motivation and weight down. The other thing is just to break that habit of reaching for food all the time, […] just to break that eating habit really.”*



*D4 “I’ve got to say, all my most memorable weight loss patients are the ones that have been on some kind of VLED and they’ve lost in excess of 20 kgs, 20, 30, 40, 50 kgs—I’ve never been able to achieve that kind of weight loss with just telling someone to cut back on their food.”*


#### 3.2.3. Experience Informed VLED Use

The participants reported that their confidence with meal replacements and their positive regard towards their use was fostered in experiences they had whilst working as dietitians. They reported that prior to these experiences they had not seriously considered MRPs or VLEDs as a valid weight loss management intervention option outside of a pre-surgery population.


*D3 “I mean it was a huge learning curve because it took me a long time to get my head around the fact that this was such an extreme measure but you know the results that we were seeing were really, really positive and the outcomes were really good, so I guess, you know that feeds back in, you know as a dietitian you know, you’re a scientist and so it’s nice to see those results.”*


#### 3.2.4. VLED Requires Long Term Involvement

Despite their confidence in using an intensive VLED (<800 cals/day), they were rarely the first line of treatment offered. The dietitians were aware they needed to be able to offer long-term, intensive support during and after a VLED and this lack of assured long-term follow-up in private practice was cited as a major barrier to utilising an intensive VLED as an intervention. Without this support there was a fear that clients would have rebound weight gain and return to old habits.


*D13 “I find most people who use shakes go back to their normal way of eating. [..] And by that time (going back to food) my EPCs (Enhanced Primary Care now known as CDM) have run out, and they may not or cannot continue as a private patient and I just can’t help them any further, it’s just sad. It’s one of the reasons why I decided not to practice as much anymore, because it was getting me down.”*


#### 3.2.5. Systemic Barriers Constrain Effective Dietetic Practice

The dietitians in this study revealed several key obstacles that interacted with each other in a way that made intervening in obesity and overweight, including the use of VLEDs and MRPs, frustrating. The participants felt that their expertise and experience made them confident, qualified and well placed in effecting positive change in Australia’s obesity epidemic, but that structures and processes outside of their control hindered progress in this area. Their frustration stemmed from recognising that they were being constrained from delivering Evidence Based Practice (EBP) in a large proportion of their clients.


*D9 “If we have evidence-based guidelines for overweight and obesity, which we do and we have frequency of contact that the evidence-based guidelines set out, which we do—would be to assess whether we can actually get that working in a public or even a private setting. You know, if we only have Medicare subsidisation for 5 dietetic sessions a year, how do we fit that in around all the evidence-based guidelines—I don’t think we can have Australian evidence-based guidelines and Australian funding models which aren’t consistent. We’ve got to try and streamline that, because how can people practice evidence-based guidelines otherwise?”*


### 3.3. Medicare and CDM Scheme Limit Effective Dietetic Care

General Practitioner Management Plans (GPMPs) for Chronic Disease Management (CDM) presented insurmountable obstacles to intervening in chronic diseases, especially those for which weight loss is a recognised evidence-based treatment. The most frustrating obstacles that were induced by this scheme were reported as: (1) insufficient rebate, (2) insufficient time allocated to consultations and (3) insufficient number of consultations. These three factors are detailed below.

#### 3.3.1. Insufficient Rebate through Medicare Could Cause Cutting Corners

The dietitians participating in this research who had experience with the CDM Medicare rebate referral scheme felt pressured to bulk bill their patients. This pressure came from the referring GPs and from the patients themselves. Dietitians felt tension between helping people and being adequately remunerated. The complexity of treating obesity causing chronic diseases and the in-depth assessment needed to formulate a management plan that suited a client’s lifestyle and skill set, meant dietitians had no choice but to charge a gap fee to provide adequate remuneration for the time of consultation not covered by Medicare rebates.


*D14 “In theory it (Medicare rebate scheme) should open up more possibilities, in theory it should open up more patients being able to come, but if we’re going to bulk bill them, it needs to be with a proper time frame and a proper payment, but the current system is just very messy and it doesn’t work. Medicare hasn’t done anything good for the profession sadly.”*


#### 3.3.2. Length of Consult Is Insufficient to Adequately Address Chronic Disease

The length of consult for the Medicare rebate is 20 min. Dietitians faced an ethical dilemma between giving what they felt was a thorough assessment and treatment strategy and being accessible to people with lower incomes. The dietitians who charged a gap fee felt that cutting corners in their service by limiting consultations to 20 min would not only hurt their own reputation but the entire dietetic profession. Those dietitians who did bulk bill, and thus remained accessible to lower income earners, reported that they spent less time with their bulk billed patients.


*D16 “What can you do in 20 min, really? A follow up maybe but you know even for that, especially if you’re doing a lifestyle intervention—you just can’t do that in 20 min. You have to build rapport to actually get some results and it’s just impossible to do.”*


#### 3.3.3. Number of Visits through Medicare Does Not Ensure Adequate Follow Up

The Medicare CDM/EPC scheme offers between 1 and 5 visits per year with a dietitian at the GP’s discretion. Dietitians tried to balance helping patients to make small permanent changes over time and needing to impart as much information as they could in the short time they had. To overcome the deleterious effect of working on long term chronic illnesses in the short-term way that Medicare allowed, some of the dietitians in this sample reported offering cheaper ‘top up’ appointments, free emailing and phone calls after hours, to their clients in order to extend support. This had a further negative effect on the morale of the dietitians who reported wanting to provide follow up care but were unable to due to a lack of remuneration.


*D13 “If I’ve got 3 to 5 visits, I try and spread them out as much as I can without affecting what I think will be progress for them, you know if they’re spread out too much it doesn’t work either.” … “It’s just these people you’re seeing for short periods of time that you won’t see again until next year when they get another EPC and they’re back to where they were when you saw them for the very first time and that can be pretty demotivating for a practitioner, let alone for the patient themselves.”*


### 3.4. Access to Health Information Makes Assessment and Treatment Goals Difficult

Often referrals from GPs arrived without pertinent health information, including pathology results and medication regimes. The dietitians that did not have access to the practice software also noted that often, they were ‘flying blind’ without their patients’ information and that further time was carved out of their allocation in chasing pertinent information for their clients. The inability to be able to either order or even access pathology results without going through a GP was cited by dietitians as a major hurdle in assessing their clients’ current health status and their progress over time. Dietitians perceived that addressing this concern/issue would enable them to provide a more targeted intervention and save them valuable time and money. Some dietitians, who had a close working relationship with the GP, would send clients back to have another appointment with the GP to request specific blood tests. This represented more lost time for the practitioners and the patients, used up a valuable appointment with the dietitian and effectively added another unnecessary cost to the Medicare system. Only one dietitian mentioned MyHealthCare, which is a government initiated online system designed to integrate a persons’ health information and make it available to them anywhere, at any time.


*D11 “It would be wonderful (to have pathology), yeah, I mean that’s part of the reason why I’m weighing, is to see what kind of progress we’re making. Without having access to the pathology with the blood sugar or the cholesterol it’s hard to tell whether the weight loss is even having any difference to long term health. […] Having access to it would be really valuable.”*


The aforementioned barriers that dietitians faced meant that in many cases dietitians could not gauge the effectiveness of their treatments.

### 3.5. Successful Outcomes Are Predicated on Working Outside of Systemic Barriers

Dietitians felt fulfilled in their practice and proud of their patients when together, they had successfully affected positive health changes through diet and lifestyle modification. Regardless of the lifestyle modification and weight loss intervention used, the most successful outcomes reported were in patients who paid privately for their care and stayed in care long term. A large proportion of the successful outcomes were achieved by using VLEDs followed by a long follow up protocol of reintroducing food in an individualised, sustainable way. These positive outcomes proved to the dietitians that they could effect change, given the necessary time.


*D11 “The person I was talking to you about just now who was really successful, I’ve seen her since January I think 6 times and that’s the kind of follow up you need, 6 times in 4 months and you know for the rest of the year she will see me less often because she’s doing great. So, if there was some sort of obesity programme where you can get 10 visits to a dietitian for a year I think that would make a really big difference, but I don’t know if there is scope for that, probably not.”*


## 4. Discussion

The descriptive themes found in this research show that dietitians prefer patient-centred care when treating obesity and chronic disease in private practice, and that they incorporate VLEDs and MPRs as a treatment option in specific situations where the settings are right for providing extended support. Overall, dietitians in this study were not averse to using MPRs and VLEDs in practice; however, they faced systemic barriers imposed by the health system which constrained dietitians from applying evidence-based practice (EBP). Additionally, successful obesity intervention was possible but was predicated on working outside of the Medicare system.

Our results demonstrate that, overall, dietitians in private practice prefer individualised, mutually agreed upon treatment protocols and goal setting. Patient-centred care is the cornerstone of high quality healthcare and is based on building rapport, fostering trust, understanding the needs of the patient and working together to formulate a treatment plan [41]. A 2017 review of dietetic practice showed that patients who felt a good rapport with their dietitian were more motivated, felt encouraged and were more likely to share important information [42]. It was evident that the dietitians in this study recognised this and actively tried to foster rapport and mutual respect with their patients.

Evidence indicates that VLEDs and MPRs are effective ways to induce weight loss [17,43] and the dietitians in this study felt they were an important part of their treatment ‘tool box’ and would offer them as a treatment option to their patients. They described that MRPs and VLEDs could help motivate their patient to continue with weight loss efforts. It was evident that the dietitians in this study had built their confidence in using VLEDs through positive experiences and training throughout their careers. Consequently, the aspect of lack of confidence in using VLEDs shown in a recent survey was not evident in this cohort [27]. One dietitian in our sample, who was the only one without practical experience in using MRPs or VLEDs, was opposed to their use outside of pre-surgical patients. This appears to support the recent finding that health care practitioners who had formal training and understood the evidence for using MRPs and VLEDs for weight loss, were more likely to prescribe MRPs or VLEDs to their patients [27]. It is not known what type of training or VLED related competency is required in Dietetic training courses.

In a service-driven discipline such as dietetics, the need for time is central to effective practice. Barriers to VLED use and other evidence-based practices for obesity treatment are related to a lack of time for the return remuneration that is possible through the current publicly-funded (Medicare) scheme. At present dietitians can only treat patients with overweight/obesity within the Medicare system after they have developed one or more long standing chronic conditions and had complex care needs. These constraints appear to negatively impact dietetic practice and limit potential for effective obesity-related chronic disease care. Dietitians acutely experience the tension between providing the time needed to deliver best practice and obtaining adequate remuneration. This tension has been identified by Allied Health Professionals Australia who contend that the “current fees [for the Medicare rebate] remain well below the true cost of providing adequate patient care and are limiting access for many of those with the greatest need.” [41]. In addition, a qualitative study exploring Australian private practicing dietitians understanding of efficiency and effectiveness reported that finding a balance between efficiency and effectiveness was difficult when working in small business or as a contractor [44]. This paper posited that increased use of health technology may help streamline administration duties that curtailed private practitioners’ billable hours [44]. Lack of time, secondary to inadequate remuneration for time in dietetic practice, has previously been shown to hinder patient-centred care [42], the use of VLEDs [27], dietary assessment [45], and best practice in obesity management [25]. Regardless of the weight loss method, overweight and obesity treatment needs to be comprehensive and requires ongoing, regular monitoring and management that continues well past the initial weight loss period if weight loss is to be maintained in the long term [24,28,46,47,48,49]. This paucity of time and the increasing pressure on the healthcare system also adversely affects the transfer of knowledge between healthcare providers.

Efficient and easy to use referral systems that ensure adequate knowledge transfer of patient history and any current test results are essential to provide evidence-based care and assess outcomes. The referral pathway towards a CDM referral requires the GP to prepare a GP Management Plan (GPMP) and Team Care Arrangements (TCA) (Appendix A) [9]. Research has shown GPs report time constraints that limit their use of the Medicare rebate scheme for chronic disease management [50]. The convoluted pathway to referral may explain why the dietitians in this study consistently contended that a lack of health information on the patient from the GP hindered their ability to properly assess and monitor their patients’ progress. To help bridge this information gap, MyHealthRecord, an Australian Government initiative, was to provide easier sharing of information between patients and multiple health care providers. It is designed to collate a patients’ health information and make it available to them to access whenever and wherever they need it. However, this system is again incumbent upon GPs to upload their patients’ files and, according to The Royal Australian College of General Practitioners (RACGP), this places an enormous burden on GPs to curate yet another set of patient files [51]. The MyHealthRecord system is also incumbent on the patient knowing about it, knowing how to gain access and having the ability to be online. Research has shown that community awareness of and GP interest in MyHealthRecord is lacking [52], and a recent study found the system has usability issues that could negatively affect certain sectors of the community [53]. It appears that the dietitians in this study are also either not aware of MyHealthRecord or their patients were unaware of it, had not set up access or GPs had not added any information to the system. Consequently, for the dietitians in this study who were not attached to a practice management system, this created a situation that make it difficult for them to assess their patients’ needs and to monitor their progress. This would indicate that facilitating awareness and access to time-saving technologies should be prioritised to ensure maximisation of knowledge exchange between patients’ healthcare providers.

Dietitians are the only healthcare providers who are professionally trained to deliver evidence based, individualised medical nutrition therapies combined with validated behaviour change techniques. Outside of government funded schemes, the dietitians in this study reported many fulfilling and successful outcomes with their patients. These dietitians spoke at length of the different strategies they had used to bring their patients through a long and difficult process to lose weight and improve their health in a sustainable way. Regardless of the strategies used, the unifying aspect that allowed them to do this was being able to deliver an individualised treatment plan in longer consultations over multiple contact points for an extended period from months to years. Conversely, a study performed with patients in hospital and community settings, solely government funded, previously reported that due to lack of length or frequency of support or the continuity of care of the dietitian, the support felt limited [54]. This is unlikely to improve, given that government-funded, public health obesity services in Australia are becoming less individualised, inflexible and more group focussed, with growing waiting times. Given these conditions, dietitians in primary care seem best placed to deliver appropriate care to the growing number of people with obesity in the population. Unfortunately, the successful treatment described by the dietitians in this study was reliant on patients paying for their own care, and this would indicate that those who cannot afford it are left without adequate care.

There is a growing disparity in the healthcare system in Australia, and obesity disproportionately affects individuals of lower socioeconomic status, the majority of whom do not have private health insurance [55,56]. Avenues for effective obesity treatment for persons with lower socioeconomic status appear limited. Other evidence-based models of care need testing to provide robust direction for government funding. In 2016, the Ministry of Health set up a taskforce to review Australia’s MBS scheme and called for stakeholder input. During this process, the AAHP and Dietitians Australia (DA) submitted a number of recommendations that, between them, addressed most of the systemic issues that are found in this study [57,58,59]. In 2019, the task force released their report and made recommendations across numerous health areas, including allied health. None of their recommendations addressed the issues found in this study directly, but there was an acknowledgment that initial consultations should be extended to 40 min, communication between healthcare providers be improved, and it was concluded that more research is needed [60]. None of the MBS task force recommendations for Allied Health have been actioned and there is no stipulation in the report for how, when, or who will carry out the further research that is needed prior to any changes being made [60]. At present there is little research on the effectiveness of allied health practitioners. Research in the private space is challenging and data difficult to collect [61]. The reported successful treatment in patients from participants for whom evidence-based practice was possible in this study, and the recommendations by the MBS task force, suggests a priority area for research.

Although limited, research to date suggests that dietetic care is effective in delivering clinically significant outcomes; however, none have taken place in private practice. In 2019, a meta-analysis of 14 worldwide randomised controlled trials showed that those in dietetic care lost an additional 1.03 kg (95% CI:−1.40; −0.66, *p* < 0.0001) compared with usual care (GP, nurse, handout) [62]. However, only two of the studies included in that meta-analysis were from Australia, in 1995 and 2006, with one having an active intervention of only 8 weeks and the other 6 months [63,64]. Another systematic review of randomised controlled trials demonstrated that in 18 out of 26 studies dietetic intervention showed statistically significant positive results across at least one measure versus the comparator. These measures included diet quality, anthropomorphic measures and clinical indicators of metabolic health (e.g., blood pressure, glucose levels, blood lipids) [65]. The dietitians in this study also reported seeing these successful health outcomes when providing evidence based long term support and they could measure the outcomes of their patients. This indicates that increased utilisation of dietitians with targeted funding models could help make positive changes to reduce obesity and the related chronic diseases and ease the pressure that is currently being felt by the healthcare system in Australia.

## 5. Strengths and Limitations

A limitation of this study is the lack of generalisability to a whole population. While private practicing Dietitians who function within the MBS, CDM government funding scheme are likely to share experiences similar to those in our study, it is not known whether this sample is more experienced in obesity treatment or whether they are more predisposed to VLED use than other Dietitians not included in our sample. VLEDs are sometimes seen as contentious in dietetics and this study may have attracted dietitians who were particularly interested and knowledgeable about this topic. Strengths of this study include the detailed and nuanced information from professionals with many years of experience in the dietetic field, which was able to provide in-depth knowledge of private practice in Australia. This body of work is unique to Australia; however, some findings may be generalised to other dietitians in this space. Qualitative research is used to gather rich in-depth knowledge of phenomena and as such, generates hypotheses that may be tested quantitively in the future. As this is the first work and exploratory, it is novel and an important contribution to the literature. It is feasible that this research could be used as a basis for studies in other countries, as pertaining to their particular healthcare system.

## 6. Conclusions

In conclusion, VLEDs are valuable tools in the treatment of obesity. However, the current government funding model, intended to extend care for those with long-term chronic conditions, creates barriers to delivering effective care using VLEDs (or any other type of dietetic intervention for obesity) and measuring outcomes. Additionally, dietitians in private practice are a knowledgeable, under paid and under-utilised resource that could ease the burden of acute care services by providing EBP for the treatment of obesity related chronic disease. Dietitians in this study were confident in their knowledge and skills, which included the use of VLEDs and MRPs, and reported successful outcomes outside of the current systemic structures that make it difficult to deliver long term, complex, evidence-based management. It appears clear that obesity, as a national health crisis, could be treated more effectively for those most affected if barriers to care could be eradicated. Addressing time constraints through adjustments to rebate schemes and health information transfer between GPs and dietitians could translate to treating more people effectively in private practice. Furthermore, the research needed to implement the recommendations made to and by the MBS task force needs to be specified by them and fast tracked, to halt the growing inequity in the current healthcare system and improve the health of the nation.

## Figures and Tables

**Table 1 healthcare-10-00404-t001:** Dietitians and years and areas of experience.

Dietitian	Years of Experience	Private Practice and/or GP or Other Specialist Practice	Private Bariatric Surgery	Public Health, Community, Hospital or Private Hospital	Research	Commercial, Industry and/or Health Fund Programme
1	5	✓		✓		✓
2	7	✓	✓		✓	
3	18	✓	✓	✓		
4	30	✓		✓	✓	
5	51	✓		✓	✓	
6	43	✓		✓	✓	
7	9	✓	✓	✓		
8	13	✓	✓	✓		
9	19	✓		✓	✓	
10	25	✓		✓		
11	4	✓	✓			
12	4	✓		✓	✓	
13	10	✓		✓	✓	
14	25	✓	✓	✓		
15	24	✓		✓		✓
16	10	✓	✓	✓		
17	28	✓				
18	25	✓		✓	✓	✓
19	15	✓				
20	20	✓		✓	✓	

**Table 2 healthcare-10-00404-t002:** Structure of major themes and explanatory subthemes.

Major Themes	Explanatory Subthemes
1. Patient centred care is dietitians’ preferred intervention model	
2. VLEDs promote weight loss in specific situations.	2.1 Experience informed VLED use and 2.2 VLED requires long term involvement.
3. Systemic barriers that constrain effective dietetic practice	3.1 Medicare and CDM scheme limits effective dietetic care. 3.1.1 Insufficient rebate through Medicare could cause cutting corners 3.1.2 Length of consult is insufficient to adequately address chronic 3.1.3 Number of visits through Medicare does not ensure adequate follow up.
	3.2 Poor access to health information makes assessment and treatment goals difficult.
4. Successful outcomes are predicated on working outside of systemic barriers	

## Data Availability

The data presented in this study are available on request from the authors.

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
