# Peer review of "“Dietitians May Only Have One Chance”—The Realities of Treating Obesity in Private Practice in Australia"

_healthcare, 2022, doi:10.3390/healthcare10020404_

Round 1

Reviewer 1 Report

“Dietitians may only have one chance” – the realities of treating obesity in private practice in Australia.

In this study authors used the descriptive qualitative methods to explore accredited practicing dietitians’ (APDs’) perspectives and practices regarding obesity and obesity interventions, including the use of very low energy diets (VLEDs) and meal replacement products (MRPs). Qualitative in-depth semi-structured interviews were conducted with 20 dietitians who had experience in private practice and in treating obesity. Transcribed interviews were analysed thematically using the technique of template analysis. 4 overarching problems were found. Ultimately, the authors considered the dietitians in private practice are well placed and able to provide life-enhancing and evidence-based treatments for overweight and obesity and the associated chronic diseases, in the community. However, systemic barriers need to be addressed, to provide equitable access to effective care irrespective of socio-economic status. Also, the descriptive themes found in this research show that dietitians prefer patient centred care when treating obesity and chronic disease in private practice, and that they incorporate VLEDs and MPRs as a treatment option in specific situations where the settings are right for providing extended support.

The work is very interesting. In a simple way presents the timeless problems of ordinary people in a big country. The authors approached the topic in a very honest way. They chose several possible paths, establishing cooperation with dietitians, and then authors conducted very extensive interviews that touched upon a large number of very important issues. 4 overarching themes were described in Table 2 and supplementary material. Very good group of respondents (Table 1) – among 20 dietitians, only three had average experience (3-4 years) in the research topic.

Many aspects of the introduction of VLEDs and MRPs as a therapeutic therapy have been carefully examined. The interpretation of the obtained data was presented in a very clear way, the authors summarize each of the discussed issues / treatment limitations, providing an appropriate justification (quotation of a dietitian). The authors discuss solutions to the problems, and also provide options for solving problems on the line of insurer-attending physician-dietitian. According to authors “dietitians are the only healthcare providers who are professionally trained to deliver evidence based, individualised medical. Strengths of this study include the detailed and nuanced information from professionals with many years of experience in the dietetic field.

I have only one objection: since Table 1 is in the main document it should not be in supplementary material.

Reviewer 2 Report

Harper et al. present a very interesting report on the management of obesity by dietitians in private practice in Australia. However, to increase the usage of the data shown in this manuscript, it would be advisable to make some changes to improve its quality.

it is advisable to shorten the introduction and the use of abbreviations

It is necessary to establish the sample size and state the size of the universe of dietitians in Australia.

A significant fact to highlight in this work is artificial intelligence to process the information coming from the interviews. The authors must underline this point in the title, the abstract, and the discussion. 

Although the information presented in this manuscript is important, if it is not compared with previous data or projections, it becomes only a report, which could be published in a letter to the editor or a brief report. Therefore, it would be highly recommended that the authors propose a new version of this manuscript emphasizing the transnationality of the information presented and elaborate a plan to take advantage of the information generated. 

Round 2

Reviewer 2 Report

none